# QUANTIFYING UNCERTAINTY WITH GAN-BASED PRIORS

## ABSTRACT

Bayesian inference is used extensively to quantify the uncertainty in an inferred field given the measurement of a related field when the two are linked by a mathematical model. Despite its many applications, Bayesian inference faces challenges when inferring fields that have discrete representations of large dimension, and/or have prior distributions that are difficult to characterize mathematically. In this work we demonstrate how the approximate distribution learned by a generative adversarial network (GAN) may be used as a prior in a Bayesian update to address both these challenges. We demonstrate the efficacy of this approach by inferring and quantifying uncertainty in inference problems arising in computer vision and physics-based applications. In both instances we highlight the role of computing uncertainty in providing a measure of confidence in the solution, and in designing successive measurements to improve this confidence.

## 1 INTRODUCTION

Quantifying uncertainty in an inference problem amounts to making a prediction and quantifying the confidence in that prediction. In the context of an image recovery problem, this may be understood as follows. A typical computer vision algorithm uses a noisy version of an image and prior knowledge to produce the recovered image which can be interpreted as the "best guess" of the original image. Quantifying uncertainty in this context involves generating an estimate of the level of confidence in the best guess, in addition to the guess itself.

Bayesian inference provides a principled approach for quantifying uncertainty. As shown in the following section, it treats the inferred vector as a multivariate stochastic vector and leads to an expression for its distribution. This expression can be used to estimate the most likely solution (the maximum a-posteriori estimate, or the MAP), the mean, the variance, or any other population parameter of interest. Thus providing a recipe for thoroughly quantifying the uncertainty in an inference problem. For the image recovery problems considered in this paper, Bayesian inference not only provides the best guess of the true image, but also a means to estimate measures of uncertainty such as the pixel-wise variance.

The knowledge of uncertainty in a prediction can directly influence the downstream action that depends on the inference. Consider an image recovery problem where two distinct inputs lead to similar recovered images: those of a traffic sign with a high speed limit. However, for the first input the predicted variance is small, while for the second input it is large. Further, the set of likely images in the second set also includes images of a Stop Sign. Then the appropriate action for the two inputs, determined after solving the inference problem and quantifying uncertainty, is very different. For the first input, the appropriate action is one of continued motion, whereas for the second input it is to slow down. Similar examples can be drawn from other areas as well, like medical imaging, high frequency trading, and autonomous systems where critical decisions are made based on the output of AI system (Gal (2016); Begoli et al. (2019)).

The knowledge of uncertainty can also be useful in determining the optimal location of a sensor. Consider an image recovery problem, where the goal is to infer the signal, and associated uncertainty, using limited amount of measurement data. In this problem a user can leverage information about the spatial distribution of uncertainty to choose the location with maximum uncertainty as next measurement location. This task falls within the fields of active learning and/or design of ex-

periments (DeGroot et al. (1962); Houlsby et al. (2011)) and is particularly useful in applications like satellite imaging, where each measurement requires significant time and/or resources.

In Figure 1, we demonstrate how the proposed GAN-based Bayesian inference algorithm is useful in both scenarios explained above: (i) in quantifying uncertainty - "*pixel-wise variance*" - our quantitative measure of uncertainty in the inferred field, and (ii) in determining the optimal sensor placement location in an iterative fashion. We return to these applications in greater detail in Section 4.

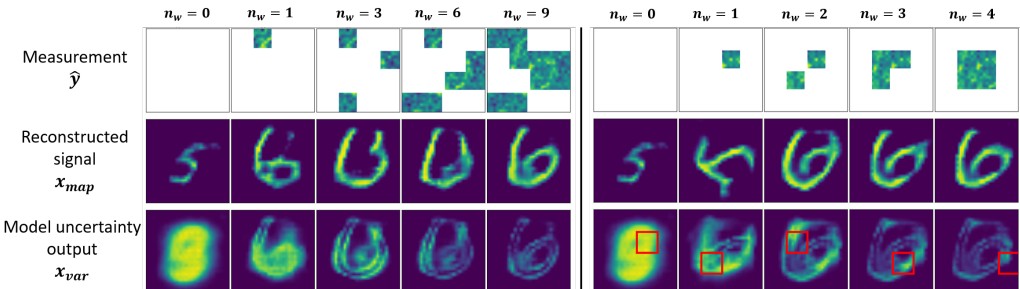

Figure 1: Estimate of the MAP (2nd row) and pixel-wise variance (3rd row) from the limited view of a noisy image (1st row) using the proposed method. The first five columns correspond to a strategy where the next window is selected randomly, while the other columns correspond to a strategy where the next window is selected in the region with maximum estimated variance. For equivalent accuracy, the variance-driven selection strategy uses fewer sampling windows (4 versus 9). In both cases variance reduces with increasing measurement.

## 1.1 BAYESIAN INFERENCE

Bayesian inference is a well-established technique for quantifying uncertainties in inference problems (Kaipio & Somersalo (2006); Dashti & Stuart (2016); Polpo et al. (2018)). It has found applications in diverse fields such as geophysics (Gouveia & Scales (1997); Malinverno (2002)), climate modeling (Jackson et al. (2004)), chemical kinetics (Najm et al. (2009)), heat conduction (Wang & Zabaras (2004)), and the detection and diagnosis of disease (Siltanen et al. (2003); Kolehmainen et al. (2006)). The two critical ingredients of a Bayesian inference problem are - an informative prior representing the prior belief about the parameters to be inferred and an efficient method for sampling from the posterior distribution. In this manuscript we describe how deep generative adversarial networks (GANs) can be effectively used in these roles.

Consider the setting where we wish to infer a vector of parameters $x \in \mathbb{R}^N$ from the measurement of a related vector $y \in \mathbb{R}^P$, where the two are related through a forward model $y = f(x)$. A noisy measurement of $y$ is denoted by $\hat{y} = f(x) + \eta$, where the vector $\eta \in \mathbb{R}^P$ represents noise. While the forward map $f$ is typically well-posed, its inverse is not, and hence to infer $x$ from the measurement $\hat{y}$ requires techniques that account for this ill-posedness. Classical techniques based on regularization tackle this ill-posedness by using additional information about the sought solution field explicitly or implicitly (Tarantola (2005)). Bayesian inference offers a different approach to this problem by modeling the unknown solution as well as the measurements as random variables. This framework addresses the ill-posedness of the inverse problem, and allows for the characterization of the uncertainty in the inferred solution.

The notion of a prior distribution plays a key role in Bayesian inference. Through multiple observations of the field $x$, denoted by the set $\mathcal{S} = \{x^{(1)}, \cdots, x^{(S)}\}$, we have some prior knowledge of $x$ that can be utilized when inferring $x$ from $\hat{y}$. This is used to build, or intuit, a prior distribution for $x$, denoted by $p_X^{\text{prior}}(x)$. Some typical examples include Gaussian process prior with specified co-variance kernels, Gaussian Markov random fields (Fahrmeir & Lang (2001)), Gaussian priors defined through differential operators (Stuart (2010)), and hierarchical Gaussian priors (Marzouk & Najm (2009); Calvetti & Somersalo (2008)). These priors promote smoothness and/or structure in the inferred solution and importantly, can be expressed explicitly in an analytical form.

Another key component of Bayesian inference is a distribution that represents the likelihood of $\boldsymbol{y}$ given an instance of $\boldsymbol{x}$, denoted by $p^{\mathrm{l}}(\boldsymbol{y}|\boldsymbol{x})$. This is often determined by the distribution of the error in the model, denoted by $p_\eta$, which captures both model and measurement errors. Given this, and an additive model for noise, the posterior distribution of $\boldsymbol{x}$, determined using Bayes' theorem after accounting for the observation $\hat{\boldsymbol{y}}$ is given by,

$$p_X^{\mathrm{post}}(\boldsymbol{x}|\boldsymbol{y}) = \frac{1}{\mathbb{Z}}p^{\mathrm{l}}(\boldsymbol{y}|\boldsymbol{x})p_X^{\mathrm{prior}}(\boldsymbol{x}) = \frac{1}{\mathbb{Z}}p_\eta(\hat{\boldsymbol{y}} - \boldsymbol{f}(\boldsymbol{x}))p_X^{\mathrm{prior}}(\boldsymbol{x}). \tag{1}$$

Here, $\mathbb{Z}$ is the prior-predictive distribution of $\boldsymbol{y}$ and ensures that the posterior integrates to one.

The posterior distribution characterizes the uncertainty in $\boldsymbol{x}$; however for vectors of large dimension characterizing this distribution explicitly is a challenging task. Consequently the expression above is used to perform tasks that are more manageable. These include determining estimates such as the maximum a-posteriori estimate (MAP), expanding the posterior distribution in terms of other distributions that are simpler to work with (Bui-Thanh et al. (2012)), or using techniques like Markov Chain Monte-Carlo (MCMC) to generate samples that are "close" to the samples generated by the true posterior distribution (Han & Carlin (2001); Parno & Marzouk (2018)).

Despite its numerous applications in solving inverse problems, Bayesian inference faces significant challenges. These include defining a reliable and informative prior distribution for $\boldsymbol{x}$ when the set $\mathcal{S} = \{\boldsymbol{x}^{(1)}, \cdots, \boldsymbol{x}^{(S)}\}$ is difficult to characterize analytically, and efficiently sampling from the posterior distribution when the dimension of $\boldsymbol{x}$ is large; a typical situation in many practical science and engineering applications.

## 1.2 RELATED WORK

The main idea developed in this paper tackles the above mentioned challenges by training a generative adversarial network (GAN) using the sample set $\mathcal{S}$, and then using the distribution learned by the GAN as the prior distribution in Bayesian inference.

The use of sample-based priors for solving an inverse problem has a rich history (Vauhkonen et al. (1997); Calvetti & Somersalo (2005)). As does the idea of reducing the dimension of the parameter space by mapping it to a lower-dimensional space (Marzouk & Najm (2009); Lieberman et al. (2010)). However, the use of learning-based deep generative models like GANs in these tasks is novel.

Recently, several authors have considered the use of learning-based methods for solving inverse problems arising in different domains. These include the use of deep convolutional neural networks (CNNs), recurrent neural networks (RNNs) or variational autoencoders (VAEs) to solve physics-driven inverse problems (Adler & Öktem (2017); Patel et al. (2019); Pesah et al. (2018)). VAEs and GANs have also been used to solve inverse problems in computer vision (Yeh et al. (2016); Ham et al.; Chang et al.; Kupyn et al. (2018); Ledig et al.; Zhu et al. (2017)). There is also a growing body of work dedicated to using GANs to learn regularizers in solving inverse problems (Lunz et al. (2018) and in compressed sensing (Bora et al. (2017; 2018); Kabkab et al. (2018); Wu et al. (2019); Shah & Hegde (2018)). However, these approaches differ from ours in that they solve the inverse problem as an optimization problem and do not rely on Bayesian inference; as a result, they add regularization in an ad-hoc manner and do not attempt to quantify the uncertainty in the inferred field.

More recently, the approach described in (Adler & Öktem (2018)) utilizes GANs in a Bayesian setting; however the GAN is trained to approximate the posterior distribution (and not the prior, as in our case), and training is done in a supervised fashion. That is, paired samples of the measurement $\hat{\boldsymbol{y}}$ and the corresponding true solution $\boldsymbol{x}$ are required. In (McCarthy et al. (2017)), authors perform variational approximation of the posterior using VAEs with simulator-based decoder to solve physics-based inference problem. We note that deep learning based Bayesian networks, where the network weights are stochastic parameters that are determined using Bayesian inference, are another means of quantifying uncertainty (MacKay (1992); Kingma & Welling (2013); Gal & Ghahramani (2016)), and have recently been applied to semantic image-segmentation and super-resolution (Kendall & Gal (2017); Kendall et al. (2019); Kohl et al. (2018); Hu et al. (2019); Tanno et al. (2019)) . However, these method also rely on supervised learning, whereas in contrast, our approach is unsupervised, and requires only samples of the true solution $\boldsymbol{x}$ to train the GAN prior.

### 1.3 OUR CONTRIBUTION

The main contribution of this paper can be summarized as follows:

1. A novel method for performing Bayesian inference involving complex priors and high dimensional posterior. In this method we utilize the distribution learned by a GAN as a surrogate for the prior distribution and reformulate the inference problem in the low-dimensional latent space of the GAN. Furthermore, we provide a theoretical analysis of the weak convergence of the posterior density learned by the proposed method to the true posterior density.

2. Novel unsupervised image denoising and inpainting algorithms with quantitative measures of uncertainty through pixel-wise variance.

3. Application of the proposed method to physics-based inference problems.

4. Demonstration of the utility of quantifying uncertainty to facilitate active learning.

## 2 PROBLEM FORMULATION

Let $\mathcal{S}$ denote the set of instances of vector $\boldsymbol{x}$ sampled from the true distribution, $p_X^{\text{true}}(\boldsymbol{x})$. Further, let $\boldsymbol{z} \sim p_Z(\boldsymbol{z})$ characterize the latent vector space and $\boldsymbol{g}(\boldsymbol{z})$ be the generator of a GAN trained using $\mathcal{S}$. Then according to Goodfellow et al. (2014), with infinite capacity and sufficient data, the generator learns the true distribution. That is,

$$p_X^{\text{gen}}(\boldsymbol{x}) = p_X^{\text{true}}(\boldsymbol{x}). \tag{2}$$

The distribution $p_X^{\text{gen}}(\boldsymbol{x})$ is defined as

$$\boldsymbol{x} \sim p_X^{\text{gen}}(\boldsymbol{x}) \Rightarrow \boldsymbol{x} = \boldsymbol{g}(\boldsymbol{z}), \boldsymbol{z} \sim p_Z(\boldsymbol{z}). \tag{3}$$

Here $p_Z$ is the multivariate distribution of the latent vector whose components are iid and typically conform to a Gaussian or a uniform distribution. The equation above implies that the GAN generates samples of $\boldsymbol{x}$ by sampling $\boldsymbol{z}$ from $p_Z$ and then passing these through the generator.

Now consider a measurement $\hat{\boldsymbol{y}}$ from which we would like to infer the posterior distribution of $\boldsymbol{x}$. For this we use (1) and set the prior distribution equal to the true distribution, that is $p_X^{\text{prior}} = p_X^{\text{true}}$. Then from (2) this is the same as $p_X^{\text{prior}} = p_X^{\text{gen}}$. Therefore,

$$p_X^{\text{post}}(\boldsymbol{x}|\boldsymbol{y}) = \frac{1}{\mathbb{Z}} p_\eta(\hat{\boldsymbol{y}} - \boldsymbol{f}(\boldsymbol{x})) p_X^{\text{gen}}(\boldsymbol{x}). \tag{4}$$

Now for any $l(\boldsymbol{x})$, we have

$$\mathbb{E}_{\boldsymbol{x} \sim p_X^{\text{post}}}[l(\boldsymbol{x})] = \frac{1}{\mathbb{Z}} \mathbb{E}_{\boldsymbol{x} \sim p_X^{\text{gen}}}[l(\boldsymbol{x}) p_\eta(\hat{\boldsymbol{y}} - \boldsymbol{f}(\boldsymbol{x}))], \qquad \text{From (4)}$$

$$= \frac{1}{\mathbb{Z}} \mathbb{E}_{\boldsymbol{z} \sim p_Z}[l(\boldsymbol{g}(\boldsymbol{z})) p_\eta(\hat{\boldsymbol{y}} - \boldsymbol{f}(\boldsymbol{g}(\boldsymbol{z})))], \qquad \text{From (3)}$$

$$= \mathbb{E}_{\boldsymbol{z} \sim p_Z^{\text{post}}}[l(\boldsymbol{g}(\boldsymbol{z}))], \tag{5}$$

where $\mathbb{E}$ is the expectation operator, and

$$p_Z^{\text{post}}(\boldsymbol{z}|\boldsymbol{y}) \equiv \frac{1}{\mathbb{Z}} p_\eta(\hat{\boldsymbol{y}} - \boldsymbol{f}(\boldsymbol{g}(\boldsymbol{z}))) p_Z(\boldsymbol{z}). \tag{6}$$

The distribution $p_Z^{\text{post}}$ is the analog of $p_X^{\text{post}}$ in the latent vector space. The measurement $\hat{\boldsymbol{y}}$ updates the prior distribution for $\boldsymbol{x}$ to the posterior distribution. Similarly, it updates the prior distribution for $\boldsymbol{z}$, $p_Z$, to the posterior distribution, $p_Z^{\text{post}}$, defined above.

Equation (5) implies that sampling from the posterior distribution of $\boldsymbol{x}$ is equivalent to sampling from the posterior distribution for $\boldsymbol{z}$ and passing the sample through the generator $\boldsymbol{g}$. That is,

$$\boldsymbol{x} \sim p_X^{\text{post}}(\boldsymbol{x}|\boldsymbol{y}) \Rightarrow \boldsymbol{x} = \boldsymbol{g}(\boldsymbol{z}), \boldsymbol{z} \sim p_Z^{\text{post}}(\boldsymbol{z}|\boldsymbol{y}). \tag{7}$$

Since the dimension of $\boldsymbol{z}$ is typically smaller than that of $\boldsymbol{x}$, this represents an efficient approach to sampling from the posterior of $\boldsymbol{x}$.

The left hand side of (5) is an expression for a population parameter of the posterior, defined by $\overline{l(\boldsymbol{x})} \equiv \mathbb{E}_{\boldsymbol{x} \sim p_X^{\text{post}}}[l(\boldsymbol{x})]$. The right hand sides of the last two lines of this equation describe how this parameter may be evaluated by sampling $\boldsymbol{z}$ (instead of $\boldsymbol{x}$) from either $p_Z$ or $p_Z^{\text{post}}$.

The equality in (5) holds for a GAN with an infinite number of weights in the generator and the discriminator. In Appendix A of this manuscript, we consider the case of a Wasserstein GAN with a finite number of weights and prove the weak convergence of the posterior density obtained by using a GAN as a prior to the true posterior density as the number of weights is increased.

## 2.1 Sampling from the posterior distribution

We consider a scenario where we wish to infer and characterize the uncertainty in the vector of parameters $\boldsymbol{x}$ from a noisy measurement of $\boldsymbol{y}$, denoted by $\hat{\boldsymbol{y}}$, where $\boldsymbol{f}$ is a known map that connects $\boldsymbol{x}$ and $\boldsymbol{y}$. And we have several prior realizations of plausible $\boldsymbol{x}$, contained in the set $\mathcal{S}$. For this problem we propose the following algorithm that accounts for the prior information in $\mathcal{S}$ and the "new" measurement $\hat{\boldsymbol{y}}$ through a Bayesian update:

1. Train a GAN with a generator $\boldsymbol{g}(\boldsymbol{z})$ on $\mathcal{S}$.
2. Sample $\boldsymbol{x}$ from $p_X^{\text{post}}(\boldsymbol{x}|\boldsymbol{y})$ given in (7).

With sufficient capacity in the GAN and with sufficient training, the posterior obtained using this algorithm will converge to the true posterior (see eq. (5) above and Appendix A). Since GANs can be used to represent complex distributions efficiently, this algorithm provides a means of including complex priors that are solely defined by samples within a Bayesian update. Further, it provides an efficient approach to sampling from the posterior since the dimension of $\boldsymbol{z}$ is typically much smaller ($10^1$ - $10^2$) than that of $\boldsymbol{x}$ ($10^4$ - $10^7$). Finally, in contrast to other methods that attempt to quantify uncertainty in image recovery tasks, this algorithm falls within the class of unsupervised learning algorithms, and does not require paired data for training. We now describe two approaches for estimating population parameters of the posterior using this algorithm.

**Monte-Carlo (MC) approximation**  The first approach is based on a Monte-Carlo approximation of a population parameter of the posterior distribution. This integral, which is defined in the second line of (5), may be approximated as,

$$\overline{l(\boldsymbol{x})} \equiv \mathbb{E}_{\boldsymbol{x} \sim p_X^{\text{post}}}[l(\boldsymbol{x})] \approx \frac{\sum_{n=1}^{N_{\text{samp}}} l(\boldsymbol{g}(\boldsymbol{z})) p_\eta(\hat{\boldsymbol{y}} - \boldsymbol{f}(\boldsymbol{g}(\boldsymbol{z})))}{\sum_{n=1}^{N_{\text{samp}}} p_\eta(\hat{\boldsymbol{y}} - \boldsymbol{f}(\boldsymbol{g}(\boldsymbol{z})))}, \qquad \boldsymbol{z} \sim p_Z(\boldsymbol{z}). \tag{8}$$

In the equation above, the numerator is obtained from a MC approximation of the integral in (5), and the denominator is obtained from a MC approximation of the scaling parameter $\mathbb{Z}$. Sampling within this approach is simple since in a typical GAN, the $z_i$s belong to a simple distribution like a Gaussian or a uniform distribution. However, we anticipate that in many applications the likelihood will tend to concentrate the distribution of latent vector $\boldsymbol{z}$ to a small region within $\Omega_z$ and thee sampling described above may be inefficient.

**Markov-Chain Monte-Carlo (MCMC) approximation**  A more efficient approach is to generate an MCMC approximation $p_Z^{\text{mcmc}}(\boldsymbol{z}|\boldsymbol{y}) \approx p_Z^{\text{post}}(\boldsymbol{z}|\boldsymbol{y})$ using the definition in (6), and thereafter sample $\boldsymbol{z}$ from this distribution. Then from the third line of (5), any desired population parameter may be approximated as

$$\overline{l(\boldsymbol{x})} \equiv \mathbb{E}_{\boldsymbol{x} \sim p_X^{\text{post}}}[l(\boldsymbol{x})] \approx \frac{1}{N_{\text{samp}}} \sum_{n=1}^{N_{\text{samp}}} l(\boldsymbol{g}(\boldsymbol{z})), \qquad \boldsymbol{z} \sim p_Z^{\text{mcmc}}(\boldsymbol{z}|\boldsymbol{y}). \tag{9}$$

We note that for all the numerical experiments in this paper we have used MCMC because of its better sample efficiency.

**Summary**  We have described two methods for probing the posterior distribution when the prior is defined by a GAN. These include an MC (8) and an MCMC estimate (9) of a given population parameter and a MAP estimate that is applicable to additive Gaussian noise with a Gaussian prior for the latent vector (see Section B in the Appendix). In the following section we apply the MCMC approach to inverse problems drawn from physics-based and computer vision applications.

## 3 EXPERIMENTS

We evaluate our method empirically on practical probabilistic field inference tasks in the domain of computer vision and physical science. In computer vision, we consider image denoising and inpainting tasks and evaluate our algorithm on MNIST and CelebA datasets. Here we attempt to infer the true image and quantify the uncertainty in this inference. We also demonstrate the utility of quantifying uncertainty by using it in an active learning setup. In the physics-based application, we consider the problem of recovering the initial temperature distribution from a measurement at later time, and use this problem to demonstrate the statistics inferred using our method converge to their "true" values.

In all cases we use a Wasserstein GAN-GP (Gulrajani et al. (2017)) to learn the prior density (architecture described in the Appendix D). We also ensure that the target images are not chosen from the set used to train the GAN. We sample from the posterior using Hamiltonian Monte Carlo (Brooks et al. (2012)) and implement it using Tensorflow-probability (Dillon et al. (2017)) library. We use initial step size of 1.0 for HMC and adapt it following (Andrieu & Thoms (2008)) based on the target acceptance probability. We use 64k samples with burn-in period of 0.5. We select these parameters to ensure convergence of chains. Using the HMC sampler we compute the MAP, which is our "best guess" of the true image, the pixel-wise mean, and the pixel-wise variance, which is our quantitative and spatially-varying estimate of uncertainty.

### 3.1 IMAGE RECOVERY USING THE MNIST & CELEBA DATABASES

We first consider the MNIST database of hand-written digits and use 55k images to train the GAN. We use a latent vector of dimension 100 with Gaussian distribution. For the image denoising task, we add Gaussian noise with zero mean and specified variance ($\sigma_y$) to the test image and use it as measurement to recover the distribution of likely images using the MCMC approach. For this problem the forward operator is the identity map, and the likelihood distribution is Gaussian. In Figure 2, we have plotted the noisy input image, the MAP estimate, and the pixel-wise mean and variance. We observe that for low and medium noise levels ($\sigma_y = 0.1, 1$), we are able to recover the original image with good accuracy, the pixel-wise variance is small overall, and is largest around the boundary of the recovered digit; this represents the variability in the different realizations of the recovered digit within the GAN prior. For the highest noise level ($\sigma_y = 10$), however, the image recovered by the MAP is incorrect in 2/3 cases, and would be misleading if viewed by itself. However, when viewed in conjunction with the estimated variance, which is large, it is clear that the confidence in the inference is small and the inferred image ought not be trusted for downstream tasks. The dependence of the average per-pixel variance in the recovered image on the variance of noise in the measured image is shown in Figure 4 (a) with 95% confidence interval. As expected, the per-pixel variance increases with increasing noise.

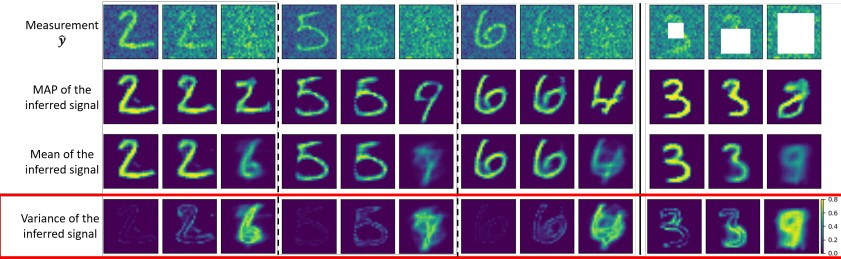

Figure 2: Estimate of the MAP, mean and pixel-wise variance from a noisy image using the proposed method. In the first three panels $\sigma_y = 0.1, 1, \& 10$, when moving from left to right. In the fourth panel $\sigma_y = 1$, and the size of the occluded region is increased.

In the right-most panel of Figure 2, we show image inpainting + denoising results for an image of digit 3 with $\sigma_y = 1$. Here the forward map is the indicator function set to zero on the occluded pixels. We note that for the small and intermediate occluded regions, the MAP solution is close to the true solution. However, when most of the image is occluded, the MAP is incorrect. Once again, the variance image, which is small for the low and medium occluded regions, and large for

the large occlusion, is a reliable indicator of the confidence in the recovered MAP image. Furthermore, the variance is peaked within the occluded region demonstrating lower confidence in the image reconstructed in this region. This is useful in applications like autonomous cars and medical imaging, where partial measurements are common and the spatial distribution of uncertainty can inform downstream tasks. More image denoising and inpainting examples are provided in Appendix C.

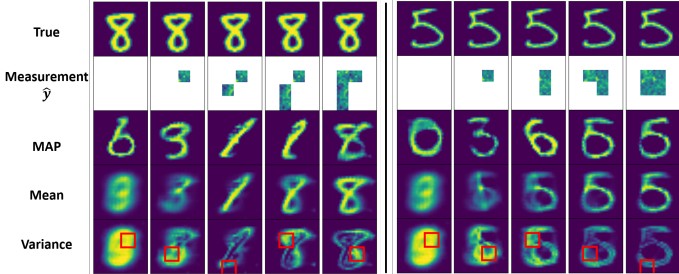

Figure 3: Estimate of the MAP, mean and variance from the limited view of a noisy image (2nd row) using the proposed method for the digits 8 & 5 (left and right panels). The window to be revealed at a given iteration (shown in red box) is selected using a variance-driven strategy.

In Figure 3, we demonstrate how uncertainty may be used in active learning/design of experiment, where the goal is to determine the optimal location for a measurement. We begin with an input where the entire image is occluded and in every subsequent step, we allow for a small $7 \times 7$ pixel window to be revealed. We select the window with the largest average pixel-wise variance. As the iterations progress, the MAP estimate converges to the true digit, and the variance decreases. In about 4 iterations we arrive at a very good guess for the digit. The performance of this approach is quantified in Figure 4(b), where we have plotted reconstruction error versus the number windows for this strategy, and a strategy where the subsequent window is selected randomly. The variance-driven strategy consistently performs better. We note that we are not aware of any other methods for computing uncertainty in recovered images that have been applied to drive an active learning task in image inpainting. While methods based on dropout (Kendall & Gal (2017); Kendall et al. (2019)) or variational inference (Kohl et al. (2018)) could be extended to accomplish this, this has not been done thus far.

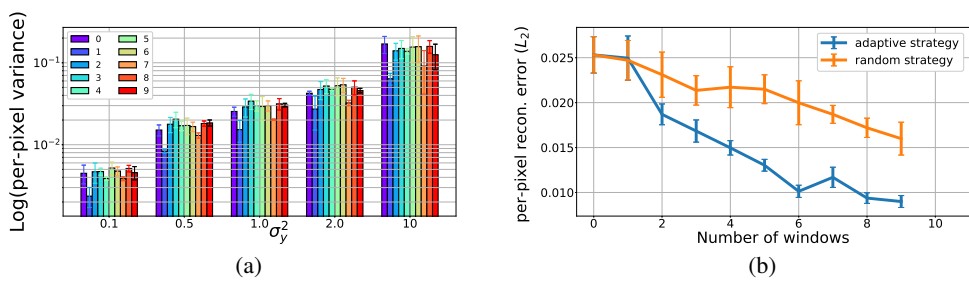

Figure 4: (a) Average variance per pixel in a reconstructed image as a function of variance in noise for 10 digits (along with 95% confidence interval). (b) Average reconstruction error (along with 95% confidence interval) as a function of number of windows for a variance-driven (adaptive learning) and a random sampling strategy.

Results for the variance-based window selection strategy applied to the CelebA dataset are shown in Figure 5. We observe that the algorithm produces realistic images at each iteration; however, the initial variance is large indicating large uncertainty. As more windows are sampled using the variance driven active learning strategy, the variance reduces and by the 7th iteration a good approximation of the true image is obtained, even though only a small, noisy portion is revealed. Additional results along with implementation details for this dataset are discussed in Appendix C.2.

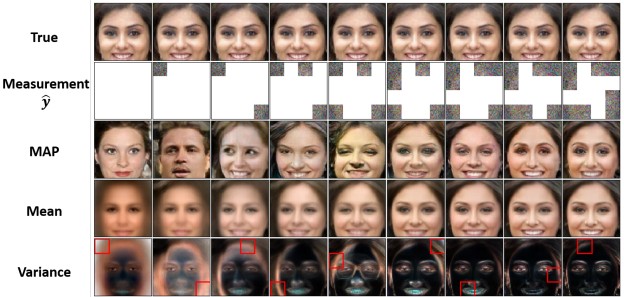

Figure 5: Estimate of the MAP, mean and variance from the limited view of a noisy image (2nd row) of a true image (1st row) using the variance-driven adaptive learning strategy.

## 3.2 A PHYSICS-DRIVEN INFERENCE PROBLEM

We consider the problem of inverse heat conduction, where the goal is to infer the initial temperature distribution (at $t = 0$) in a domain given a noisy measurement of temperature at later time ($t = 1$) and the thermal conductivity of the material. The forward map is the solution of the time-dependent heat conduction problem with uniform conductivity, $\kappa = 0.64$, and in a square domain of length $L = 2\pi$ with Dirichlet boundary conditions. This operator maps the initial temperature field to the temperature field at later time. The discrete version of this operator is obtained by discretizing the time-dependent linear heat conduction equation using the central difference scheme in space and backward Euler scheme in time. Much like a blurring kernel, the forward operator smooths the initial temperature distribution, and the extent of smoothing increases with $\kappa \times t$.

We consider a family of initial temperatures where the background is zero, and the temperature on a rectangular sub-domain varies linearly from 2 units on the left edge to 4 units on the right edge. This distribution is parameterized by four parameters, $\{\xi_i\}_{i=1}^4$, which are the coordinates of the lower left and upper right corners of the rectangular region. The sample set $\mathcal{S}$ is created by sampling each parameter from a uniform distribution ($xi_i \sim U(0.25, 0.75) \times L$) and is used to train the GAN prior. The posterior distribution is sampled using the HMC sampler.

In the top two rows of Figure 6, we have plotted the true initial condition, the noise-free temperature at $t = 1$, and the noisy temperature measurement ($\sigma_y = 1$) used as input in the GAN-based prior approach. The corresponding MAP, mean and pixel-wise variance estimated by the MCMC approximation are shown next. We observe that the MAP is very close to the true initial temperature distribution and the variance is concentrated along the edges of the rectangle where the uncertainty is the largest. In the following columns we have plotted the MAP estimate obtained assuming $L_2$ and $H^1$ Gaussian priors, which are often used to solve these types of problems, and are clearly much less accurate.

For this problem the "true" posterior can be reduced to the 4-dimensional space of parameters, and sampled by generating initial conditions corresponding to the values of these parameters. A simple MC approximation can be performed to compute statistics - the mean and the pixel-wise variance for the true posterior (last two columns of Figure 6). By comparing these with the mean and the pixel-wise variance (columns 5 & 6) estimated by the GAN-based prior, we conclude that GAN-based posterior has converged to the true posterior.

In the bottom rows of Figure 6, we plot similar results for initial conditions and GAN-based priors generated from the MNIST database. In this case the measurement is made at $t = 0.2$. Since the "true" distribution for this set is not known the true mean and variance are not plotted.

## 4 CONCLUSIONS

The ability to quantify the uncertainty in an inference problem is useful in developing confidence in that inference, and in designing strategies to improve the confidence. In this paper we have described how this may be accomplished when solving a Bayesian inference problem by using GANs as priors. Since GANs can learn complex distributions of a wide variety of fields from their

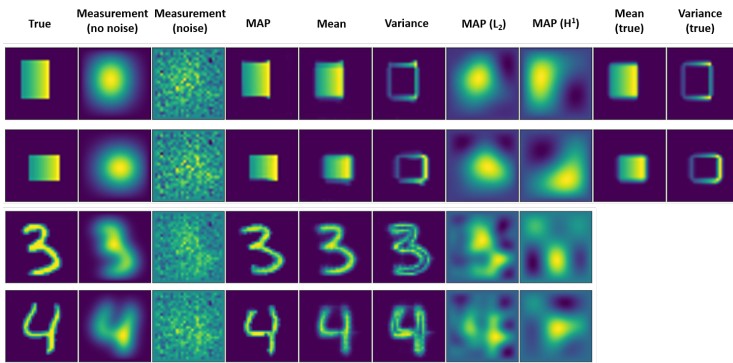

Figure 6: From left to right: (1) true initial temperature, (2) temperature at $t = 1$, (3) noisy version temperature used as measurement, (4), (5), (6) MAP, mean and pixel-wise variance estimates using GAN priors, (7) and (8) MAP estimates using $L_2$ and $H^1$ Gaussian priors, (9) & (10) true MAP and variance obtained by sampling over the true parameter space.

samples, this approach can be applied to a range of problems in computer vision and physics-driven inference. It derives its efficiency by mapping the posterior distribution to the latent space, whose dimension is often much smaller than that of the inferred field. We have applied this approach to image recovery tasks and demonstrated how the knowledge of uncertainty in the prediction can be used to assess confidence in a prediction, and via active learning to design a strategy to improve it. We have also applied this approach to a physics-based problem, where we have verified its accuracy and robustness. In the results reported in this manuscript we assume knowledge of the forward map; however, we note that the proposed algorithm can easily be extended to a regime where the forward map is also unknown by utilizing likelihood-free inference methods like ABC or meta-learning approaches. We leave this as an interesting direction for future research.

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

# A  WEAK CONVERGENCE OF THE POSTERIOR DENSITY

In this section we prove the weak convergence of the posterior density obtained by using a GAN as a prior to the true posterior density as the number of weights in the generator and discriminator are increased.

Let the generator of the GAN be given by $\boldsymbol{g}(\boldsymbol{z}; \boldsymbol{\theta})$, where $\boldsymbol{z} \in \mathbb{R}^M$ is the latent vector, and $\boldsymbol{\theta} \in \mathbb{R}^{N_\theta}$ is the vector of weights. The vector $\boldsymbol{z}$ is selected from the distribution $p_Z(\boldsymbol{z})$. Note that $\boldsymbol{g} : \mathbb{R}^M \to \mathbb{R}^N$.

Let the discriminator of the GAN be given by $d(\boldsymbol{x}; \boldsymbol{\phi})$, where $\boldsymbol{x} \in \mathbb{R}^N$, and $\boldsymbol{\phi} \in \mathbb{R}^{N_\phi}$ be the vector of weights. Note that $d : \mathbb{R}^N \to \mathbb{R}(0, 1)$.

The GAN is trained using a set of samples of $\boldsymbol{x}$, drawn from $p_X^{\text{prior}}(\boldsymbol{x})$. For the loss function, consider

$$L(\boldsymbol{\theta}, \boldsymbol{\phi}) = \mathbb{E}_{\boldsymbol{x} \sim p_X^{\text{prior}}}[\rho(1 - d(\boldsymbol{x}; \boldsymbol{\phi}))] + \mathbb{E}_{\boldsymbol{z} \sim p_Z}[\rho(d(\boldsymbol{g}(\boldsymbol{z}; \boldsymbol{\theta}); \boldsymbol{\phi}))]. \tag{10}$$

Here $\rho$ is a monotone real-valued function which defines the GAN family being analyzed. For example, for the Wasserstein GAN, $\rho(\xi) = \xi$.

The optimal values of the weights are given by

$$\boldsymbol{\theta}^*, \boldsymbol{\phi}^* = \underset{\boldsymbol{\theta}}{\text{argmax}}(\underset{\boldsymbol{\phi}}{\text{argmin}}(L(\boldsymbol{\theta}, \boldsymbol{\phi}))). \tag{11}$$

## A.1  STATIONARITY CONDITIONS

The necessary conditions for these optimal values are

$$\frac{\partial L(\boldsymbol{\theta}^*, \boldsymbol{\phi}^*)}{\partial \boldsymbol{\phi}} = \boldsymbol{0} \tag{12}$$

$$\frac{\partial L(\boldsymbol{\theta}^*, \boldsymbol{\phi}^*)}{\partial \boldsymbol{\theta}} = \boldsymbol{0}. \tag{13}$$

Using the definition of the loss function (10) in (12), we have

$$\mathbb{E}_{\boldsymbol{x} \sim p_X^{\text{prior}}}[\rho'(1 - d(\boldsymbol{x}; \boldsymbol{\phi}^*))\frac{\partial d}{\partial \boldsymbol{\phi}}(\boldsymbol{x}; \boldsymbol{\phi}^*)] = \mathbb{E}_{\boldsymbol{z} \sim p_Z}[\rho'(d(\boldsymbol{g}(\boldsymbol{z}; \boldsymbol{\theta}^*); \boldsymbol{\phi}^*))\frac{\partial d}{\partial \boldsymbol{\phi}}(\boldsymbol{g}(\boldsymbol{z}; \boldsymbol{\theta}^*); \boldsymbol{\phi}^*)]. \tag{14}$$

Similarly, using (10) in (13), we have

$$\mathbb{E}_{\boldsymbol{z} \sim p_Z}[\rho'(d(\boldsymbol{g}(\boldsymbol{z}; \boldsymbol{\theta}^*); \boldsymbol{\phi}^*))\frac{\partial d}{\partial \boldsymbol{x}}(\boldsymbol{g}(\boldsymbol{z}; \boldsymbol{\theta}^*); \boldsymbol{\phi}^*) \cdot \frac{\partial \boldsymbol{g}}{\partial \boldsymbol{\theta}}(\boldsymbol{z}; \boldsymbol{\theta}^*)] = \boldsymbol{0}. \tag{15}$$

## A.2  WASSERSTEIN GAN

For the Wasserstein GAN, $\rho(\xi) = \xi$ and $\rho'(\xi) = 1$. As a result (14) and (15) reduce to,

$$\mathbb{E}_{\boldsymbol{x} \sim p_X^{\text{prior}}}[\frac{\partial d}{\partial \boldsymbol{\phi}}(\boldsymbol{x}; \boldsymbol{\phi}^*)] = \mathbb{E}_{\boldsymbol{z} \sim p_Z}[\frac{\partial d}{\partial \boldsymbol{\phi}}(\boldsymbol{g}(\boldsymbol{z}; \boldsymbol{\theta}^*); \boldsymbol{\phi}^*)] \tag{16}$$

$$\mathbb{E}_{\boldsymbol{z} \sim p_Z}[\frac{\partial d}{\partial \boldsymbol{x}}(\boldsymbol{g}(\boldsymbol{z}; \boldsymbol{\theta}^*); \boldsymbol{\phi}^*) \cdot \frac{\partial \boldsymbol{g}}{\partial \boldsymbol{\theta}}(\boldsymbol{z}; \boldsymbol{\theta}^*)] = \boldsymbol{0}. \tag{17}$$

Let $w_a(\boldsymbol{x}) \equiv \frac{\partial d}{\partial \phi_a}(\boldsymbol{x}; \boldsymbol{\phi}^*), a = 1, \cdots, N_\phi$. Then (16) implies

$$\mathbb{E}_{\boldsymbol{x} \sim p_X^{\text{prior}}}[w_a(\boldsymbol{x})] = \mathbb{E}_{\boldsymbol{z} \sim p_Z}[w_a(\boldsymbol{g}(\boldsymbol{z}; \boldsymbol{\theta}^*))], a = 1, \cdots, N_\phi. \tag{18}$$

As $N_\phi \to \infty$, this is a weak statement of the equivalence of $p_X^{\text{prior}}$ and $p_X^{\text{gen}}$, where the latter is defined in (3). In particular this says that the push forward of the measure in the latent space under

the function $g(z)$ weakly converges to the measure associated with distribution of $x$. We note with increasing number of weights in the discriminator, the relation above is required to hold for an increasing number of test functions, $w_a$. In addition, we have implicitly assumed that the generator is rich enough, that is it has enough weights/layers, such that this relation can actually be satisfied. To make this clear consider the extreme case of a generator with a single weight; in this case there is no way that (18) will be satisfied for a large number $N_\phi$. Thus in order for this relation to hold for a large $N_\phi$, we must also provide the generator with a large $N_\theta$.

Now consider a sufficiently smooth function $m(x)$ which defines the point estimate we wish to compute. Expand it using $w_a(x)$ as basis such that

$$m(\boldsymbol{x}) - \sum_{a=1}^{N_\phi} \alpha_a w_a(\boldsymbol{x}) = \epsilon(\boldsymbol{x}), \tag{19}$$

and the coefficients are selected to minimize

$$\bar{\epsilon} = \|\epsilon(\boldsymbol{x})\|_\infty. \tag{20}$$

As we increase $N_\phi$ the dimension of the basis used to represent $m$ increases, and therefore $\bar{\epsilon}$ tends to zero.

Given this,

$$
\begin{aligned}
\mathop{\mathbb{E}}_{\boldsymbol{x}\sim p_X^{\mathrm{prior}}}[m(\boldsymbol{x})] &= \mathop{\mathbb{E}}_{\boldsymbol{x}\sim p_X^{\mathrm{prior}}}\big[\sum_{a=1}^{N_\phi} \alpha_a w_a(\boldsymbol{x}) + \epsilon(\boldsymbol{x})\big], && \text{from(19)}\\
&= \sum_{a=1}^{N_\phi} \alpha_a \mathop{\mathbb{E}}_{\boldsymbol{x}\sim p_X^{\mathrm{prior}}}[w_a(\boldsymbol{x})] + \mathop{\mathbb{E}}_{\boldsymbol{x}\sim p_X^{\mathrm{prior}}}[\epsilon(\boldsymbol{x})]\\
&= \sum_{a=1}^{N_\phi} \alpha_a \mathop{\mathbb{E}}_{\boldsymbol{z}\sim p_Z}[w_a(\boldsymbol{g}(\boldsymbol{z};\boldsymbol{\theta}^*))] + \mathop{\mathbb{E}}_{\boldsymbol{x}\sim p_X^{\mathrm{prior}}}[\epsilon(\boldsymbol{x})], && \text{from(18)}\\
&= \mathop{\mathbb{E}}_{\boldsymbol{z}\sim p_Z}\big[\sum_{a=1}^{N_\phi} \alpha_a w_a(\boldsymbol{g}(\boldsymbol{z};\boldsymbol{\theta}^*))\big] + \mathop{\mathbb{E}}_{\boldsymbol{x}\sim p_X^{\mathrm{prior}}}[\epsilon(\boldsymbol{x})]\\
&= \mathop{\mathbb{E}}_{\boldsymbol{z}\sim p_Z}[m(\boldsymbol{g}(\boldsymbol{z};\boldsymbol{\theta}^*)) - \epsilon(\boldsymbol{g}(\boldsymbol{z};\boldsymbol{\theta}^*))] + \mathop{\mathbb{E}}_{\boldsymbol{x}\sim p_X^{\mathrm{prior}}}[\epsilon(\boldsymbol{x})], && \text{from(19)}\\
&= \mathop{\mathbb{E}}_{\boldsymbol{z}\sim p_Z}[m(\boldsymbol{g}(\boldsymbol{z};\boldsymbol{\theta}^*))] - \mathop{\mathbb{E}}_{\boldsymbol{z}\sim p_Z}[\epsilon(\boldsymbol{g}(\boldsymbol{z};\boldsymbol{\theta}^*))] + \mathop{\mathbb{E}}_{\boldsymbol{x}\sim p_X^{\mathrm{prior}}}[\epsilon(\boldsymbol{x})]. && (21)
\end{aligned}
$$

This yields,

$$
\begin{aligned}
\big|\mathop{\mathbb{E}}_{\boldsymbol{x}\sim p_X^{\mathrm{prior}}}[m(\boldsymbol{x})] - \mathop{\mathbb{E}}_{\boldsymbol{z}\sim p_Z}[m(\boldsymbol{g}(\boldsymbol{z};\boldsymbol{\theta}^*))]\big| &\leq \big| - \mathop{\mathbb{E}}_{\boldsymbol{z}\sim p_Z}[\epsilon(\boldsymbol{g}(\boldsymbol{z};\boldsymbol{\theta}^*))] + \mathop{\mathbb{E}}_{\boldsymbol{x}\sim p_X^{\mathrm{prior}}}[\epsilon(\boldsymbol{x})]\big|\\
&\leq \big|\mathop{\mathbb{E}}_{\boldsymbol{z}\sim p_Z}[\epsilon(\boldsymbol{g}(\boldsymbol{z};\boldsymbol{\theta}^*))]\big| + \big|\mathop{\mathbb{E}}_{\boldsymbol{x}\sim p_X^{\mathrm{prior}}}[\epsilon(\boldsymbol{x})]\big|\\
&\leq \mathop{\mathbb{E}}_{\boldsymbol{z}\sim p_Z}[\|\epsilon(\boldsymbol{g}(\boldsymbol{z};\boldsymbol{\theta}^*))\|_\infty] + \mathop{\mathbb{E}}_{\boldsymbol{x}\sim p_X^{\mathrm{prior}}}[\|\epsilon(\boldsymbol{x})\|_\infty]\\
&\leq \mathop{\mathbb{E}}_{\boldsymbol{z}\sim p_Z}[\bar{\epsilon}] + \mathop{\mathbb{E}}_{\boldsymbol{x}\sim p_X}[\bar{\epsilon}], && \text{from(20)}\\
&= 2\bar{\epsilon}. && (22)
\end{aligned}
$$

Note that in deriving this estimate, in the third line we have assumed that $g(z;\theta^*) \in \Omega_x$. That is, the generator does not land outside the domain of $x$. The statement above bounds the error in computing a population parameter for the prior using a GAN.

## A.3 CONVERGENCE TO THE POSTERIOR DENSITY

In order to turn this estimate into an estimate for the error in approximating a population parameter for the posterior density, we simply choose $m(\boldsymbol{x}) = \frac{l(\boldsymbol{x})p_\eta(\hat{\boldsymbol{y}} - \boldsymbol{f}(\boldsymbol{x}))}{\mathbb{Z}}$. With this choice,

$$\mathop{\mathbb{E}}_{\boldsymbol{x}\sim p_X^{\mathrm{prior}}}[m(\boldsymbol{x})] = \mathop{\mathbb{E}}_{\boldsymbol{x}\sim p_X^{\mathrm{post}}}[l(\boldsymbol{x})], \qquad \text{from (1)} \tag{23}$$

and

$$\mathbb{E}_{\boldsymbol{z} \sim p_Z}\left[m(\boldsymbol{g}(\boldsymbol{z}; \boldsymbol{\theta}^*))\right] = \mathbb{E}_{\boldsymbol{z} \sim p_Z^{\text{post}}}\left[l(\boldsymbol{g}(\boldsymbol{z}; \boldsymbol{\theta}^*))\right], \qquad \text{from (6).} \tag{24}$$

Using these in (22) we have the desired result,

$$\left| \mathbb{E}_{\boldsymbol{x} \sim p_X^{\text{post}}}\left[l(\boldsymbol{x})\right] - \mathbb{E}_{\boldsymbol{z} \sim p_Z^{\text{post}}}\left[l(\boldsymbol{g}(\boldsymbol{z}; \boldsymbol{\theta}^*))\right] \right| \leq 2\bar{\epsilon}. \tag{25}$$

This result demonstrates that the difference between the true population parameter for the posterior and its approximation obtained after using a GAN as a prior (the method proposed in the paper) reduces as the number of weights in the generator and the discriminator are increased. In the limit of infinite weights, $\bar{\epsilon} \to 0$, and the equation above is a statement of weak convergence of the posterior obtained by using the GAN and the true posterior density.

## B    EXPRESSION FOR THE MAXIMUM A-POSTERIORI ESTIMATE

The techniques described in Section 2.1 focus on sampling from the posterior distribution and computing approximations to population parameters. These techniques can be applied in conjunction with any distribution used to model noise and the latent space vector; that is, any choice of $p_\eta$ (likelihood) and $p_Z$ (prior). In this section we consider the special case when Gaussian models are used for noise and the latent vector. In this case, we can derive a simple optimization algorithm to determine the maximum a-posteriori estimate (MAP) for $p_Z^{\text{post}}(\boldsymbol{z}|\boldsymbol{y})$. This point is denoted by $\boldsymbol{z}^{\text{map}}$ in the latent vector space and represents the most likely value of the latent vector in the posterior distribution. It is likely that the operation of the generator on $\boldsymbol{z}^{\text{map}}$, that is $\boldsymbol{g}(\boldsymbol{z}^{\text{map}})$, will yield a value that is close to $\boldsymbol{x}^{\text{map}}$, and may be considered as a likely solution to the inference problem.

We consider the case when the components of the latent vector are iid with a normal distribution with zero mean and unit variance. This is often the case in many typical applications of GANs. Further, we assume that the components of noise vector are defined by a normal distribution with zero mean and a covariance matrix $\boldsymbol{\Sigma}$. Using these assumptions in (6), we have

$$p_Z^{\text{post}}(\boldsymbol{z}|\boldsymbol{y}) \propto \exp\left(-\frac{1}{2} \overbrace{\left(|\boldsymbol{\Sigma}^{-1/2}(\hat{\boldsymbol{y}} - \boldsymbol{f}(\boldsymbol{g}(\boldsymbol{z})))|^2 + |\boldsymbol{z}|^2\right)}^{\equiv r(\boldsymbol{z})}\right). \tag{26}$$

The MAP estimate for this distribution is obtained by minimizing the negative of the argument of the exponential. That is

$$\boldsymbol{z}^{\text{map}} = \arg\min_{\boldsymbol{z}} r(\boldsymbol{z}). \tag{27}$$

This minimization problem may be solved using any gradient-based optimization algorithm. The input to this algorithm is the gradient of the functional $r$ with respect to $\boldsymbol{z}$, which is given by

$$\frac{\partial r}{\partial \boldsymbol{z}} = \boldsymbol{H}^T(\boldsymbol{z})\boldsymbol{\Sigma}^{-1}(\boldsymbol{f}(\boldsymbol{g}(\boldsymbol{z})) - \hat{\boldsymbol{y}}) + \boldsymbol{z}, \tag{28}$$

where the matrix $\boldsymbol{H}$ is defined as

$$\boldsymbol{H} \equiv \frac{\partial \boldsymbol{f}(\boldsymbol{g}(\boldsymbol{z}))}{\partial \boldsymbol{z}} = \frac{\partial \boldsymbol{f}}{\partial \boldsymbol{x}} \frac{\partial \boldsymbol{g}}{\partial \boldsymbol{z}}. \tag{29}$$

Here $\frac{\partial \boldsymbol{f}}{\partial \boldsymbol{x}}$ is the derivative of the forward map $\boldsymbol{f}$ with respect to its input $\boldsymbol{x}$, and $\frac{\partial \boldsymbol{g}}{\partial \boldsymbol{z}}$ is the derivative of the generator output with respect to the latent vector. In evaluating the gradient above we need to evaluate the operation of the matrices $\frac{\partial \boldsymbol{f}}{\partial \boldsymbol{x}}$ and $\frac{\partial \boldsymbol{g}}{\partial \boldsymbol{z}}$ on a vector, and not the matrices themselves. The operation of $\frac{\partial \boldsymbol{g}}{\partial \boldsymbol{z}}$ on a vector can be determined using a back-propagation algorithm with the GAN; while the operation of $\frac{\partial \boldsymbol{f}}{\partial \boldsymbol{x}}$ can be determined by making use of the adjoint of the linearization of the forward operator.

Once $\boldsymbol{z}^{\text{map}}$ is determined, one may evaluate $\boldsymbol{g}(\boldsymbol{z}^{\text{map}})$ by using the GAN generator. This represents the value of the field we wish to infer at the most likely value value of latent vector. Note that this is not the same as the MAP estimate of $p_X^{\text{post}}(\boldsymbol{x}|\boldsymbol{y})$.

## C  ADDITIONAL RESULTS

In this section we provide additional results for both MNIST and CelebA dataset for different tasks discussed in the main paper.

### C.1  MNIST

First we provide additional examples in Figure 7 for variance-based adaptive measurement window selection procedure described in Section 3.1.

Figure 8 shows additional results for the inpainting + denoising task, where MNIST digits are occluded with masks of different sizes at different locations. Note that the variance is high where the occlusion mask is located indicating lower confidence in reconstructed image in that location.

### C.2  CELEBA

For the CelebA dataset, we trained WGAN-GP model using more than 200k celebrity facial images and perform inference using remaining test set images. The input images were cropped to a $64 \times 64$ RGB image and were normalized between [-1, 1].

Once the GAN was trained, the HMC algorithm was used for posterior sampling and inference on a complimentary set of images (not used for training). In Figure 9 we show some additional results for variance-based adaptive measurement window selection procedure for CelebA dataset.

Next, in figure 10 we show some additional results for image recovery task for CelebA dataset. Once again we note that the MAP estimate and the mean is close to the true image. On the other hand, the closest image from the training set (in an $L_2$ sense) is not as accurate. This points to the utility of using the GAN as an interpolant in the latent vector space.

## D  ARCHITECTURE AND TRAINING DETAILS

We use the WGAN-GP model for learning prior density. The tuned value of hyper-parameters is shown in Table 1. We use the same generator and discriminator architecture for the MNIST and the physics-based inference problem; whereas for the CelebA dataset we use a slightly different architecture to accommodate different input image size. The layout of both these architecture is shown in Figure 11. Some notes regarding nomenclature used in Figure 11.

Table 1: Hyper-parameters for WGAN-GP model

| Parameters | MNIST | CelebA | inverse heat conduction |
|:---:|:---:|:---:|:---:|
| epochs | 1000 | 500 | 200 |
| learning rate | 0.0002 | 0.0001 | 0.0002 |
| optimizer | Adam | Adam | Adam |
| momentum params $(\beta_1, \beta_2)$ | 0.5, 0.999 | 0.5, 0.999 | 0.5, 0.999 |
| batch size | 64 | 64 | 64 |
| $n_{critic}/n_{gen}$ | 5 | 5 | 1 |

- Conv (HxWxC s=n) indicates convolutional layer with filer size of HxW and number of filters=C with stride=n.
- FC (x,y) indicates fully connected layer with x neurons in input layer and y neurons in output layer.
- BN = Batch norm, LN = Layer norm.
- TrConv = Transposed Convolution.
- LReLU = Leaky ReLU with $\alpha$=0.2.

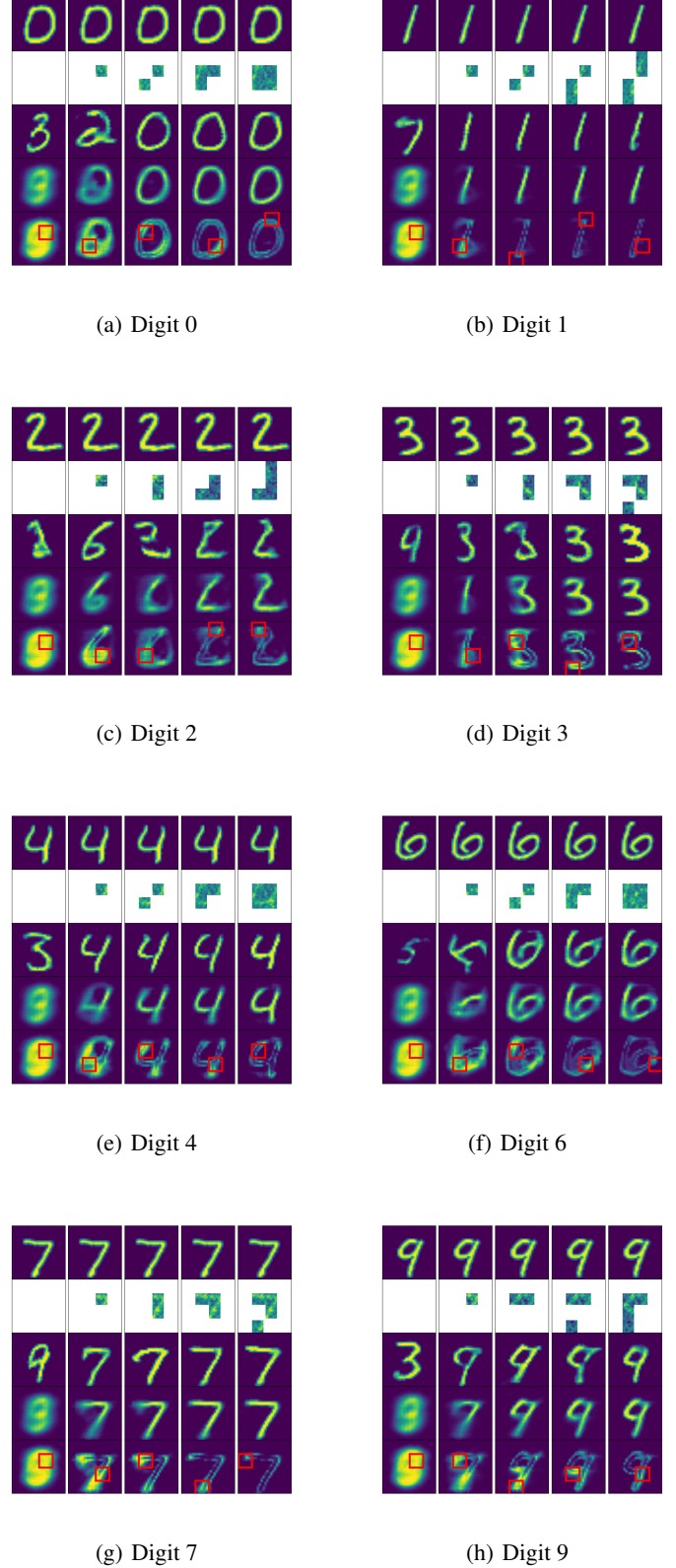

(a) Digit 0      (b) Digit 1

(c) Digit 2      (d) Digit 3

(e) Digit 4      (f) Digit 6

(g) Digit 7      (h) Digit 9

Figure 7: Estimate of the MAP (3rd row), mean (4th row) and variance (5th row) from the limited view of a noisy image (2nd row) using the proposed method. The window to be revealed at a given iteration (shown in red box) is selected using a variance-driven strategy. Top row indicates ground truth. For all digits $\sigma_y = 1$.

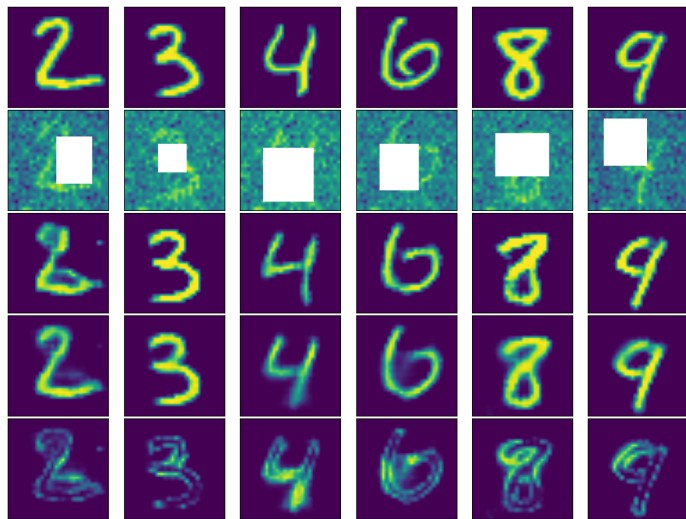

Figure 8: Estimate of the MAP (3rd row), mean (4th row) and variance (5th row) from a noisy image (2nd row) using the proposed method. Top row shows ground truth. For all the examples $\sigma_y = 1$.

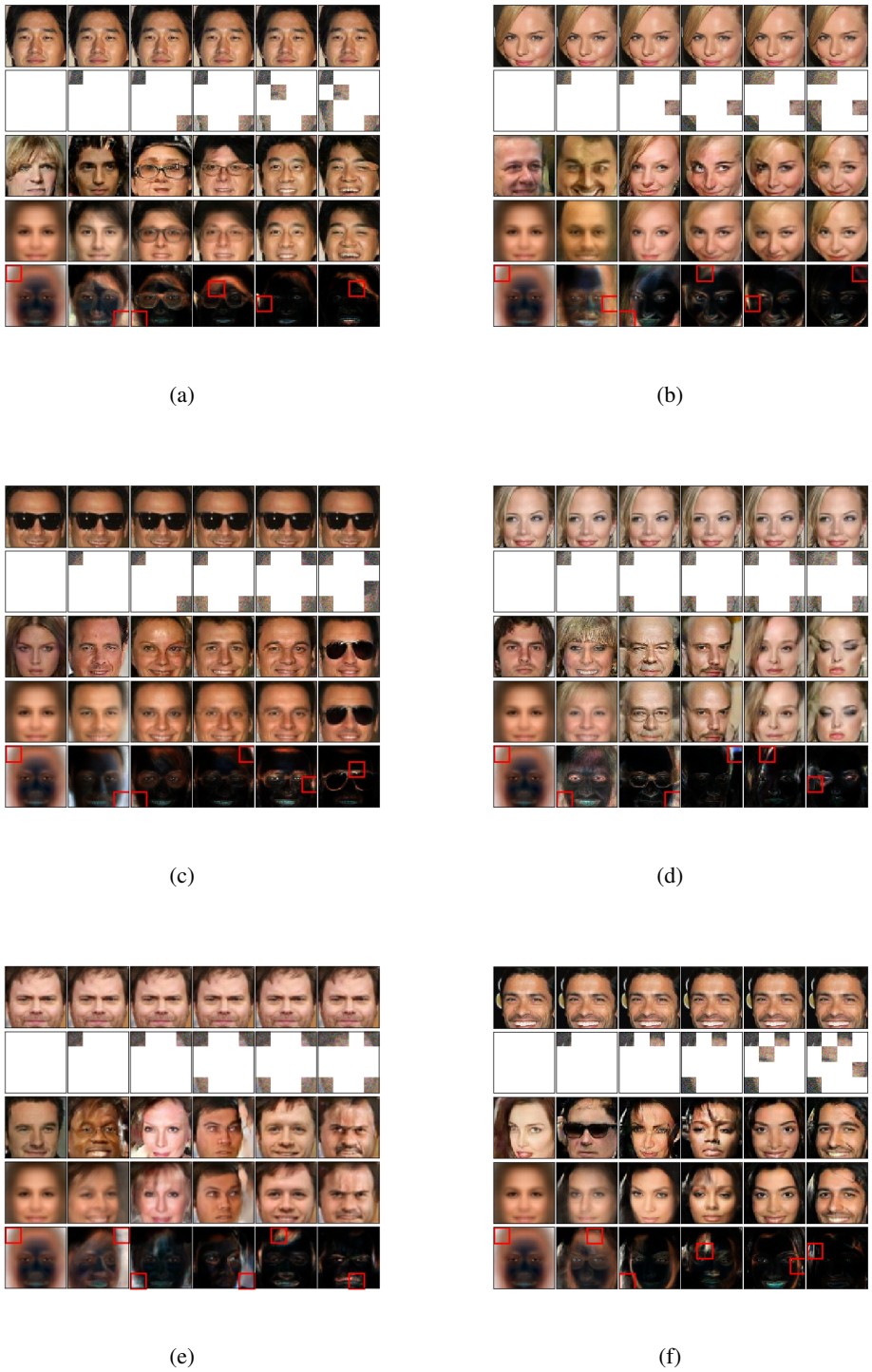

Figure 9: Estimate of the MAP (3rd row), mean (4th row) and variance (5th row) from the limited view of a noisy image (2nd row) using the proposed adaptive method. The window to be revealed at a given iteration (shown in red box) is selected using a variance-driven strategy. Top row indicates ground truth. For all images $\sigma_y = 1$.

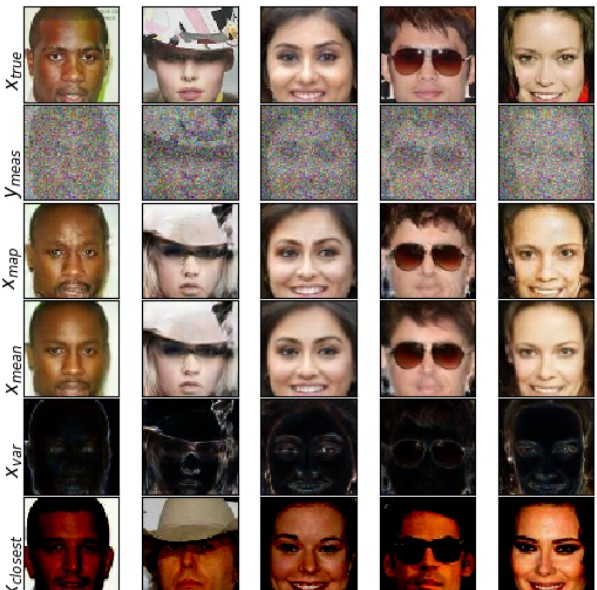

(a)

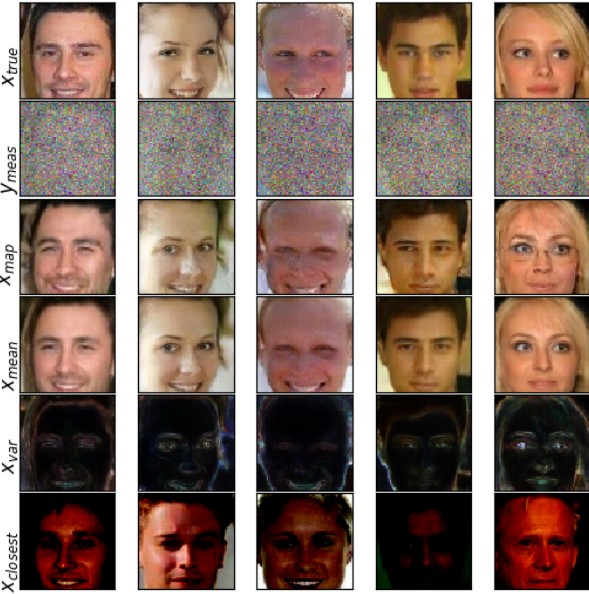

(b)

Figure 10: Estimate of the MAP (3rd row), mean (4th row) and variance (5th row) from a noisy image (2nd row) using the proposed method. Top row shows the ground truth. The last row shows the closest example in training set (by the $L_2$ measure). For all images $\sigma_y = 1$.

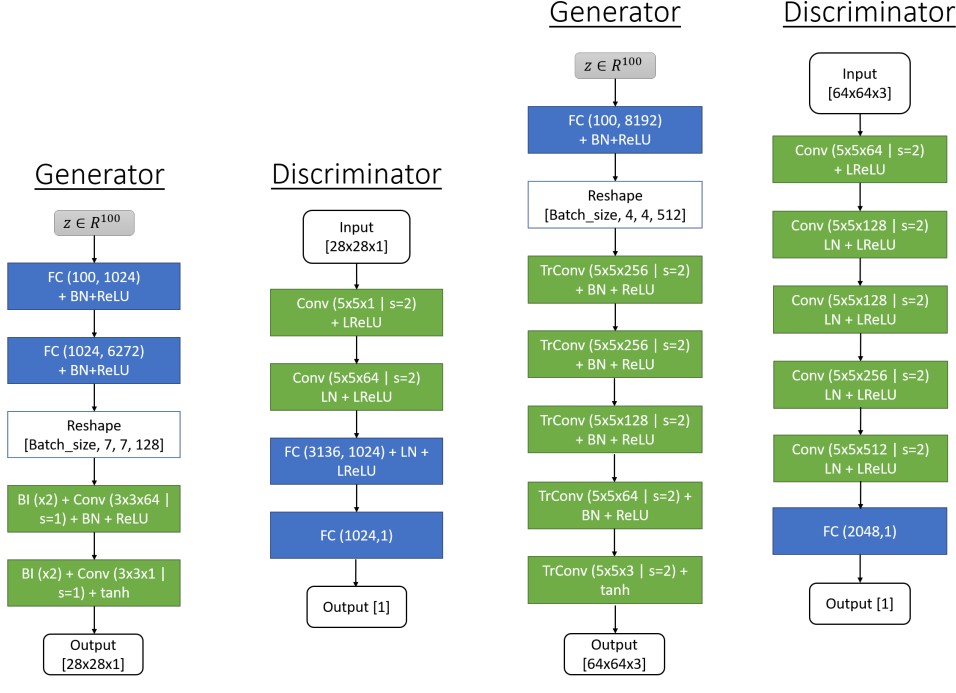

(a) Architecture for MNIST and synthetic dataset (used in physics-based inference problem)

(b) Architecture for CelebA dataset

Figure 11: Generator and discriminator architectures,

