# OpenReview forum: "Quantifying uncertainty with GAN-based priors"
_ICLR.cc/2020/Conference — Reject_

### Official Review · AnonReviewer1 · 2019-10-18
**Official Blind Review #1**

**Rating:** 3

**Review:**

Summary of the paper:

        The paper proposes a Bayesian approach to make inference about latent variables such as un-corrupted images. The prior distribution plays a key role in this task. The authors use a GAN to estimate this prior distribution. Then, standard Bayesian techniques such a Hamilton Monte Carlo are used to make inference about the latent variables.

Detailed comments:

Eq. (8) is expected to give very bad results. The reason is that it is very unlikely to sample from the prior configurations for z that are compatible with y.

The paper does not address learning any model parameters. e.g. the amount of noise.

A more principled approach would be to estimate the prior parameters using maximum likelihood estimation. That has already been done in the case of the
variational autoencoder.

The variational autoencoder is an already known method that can be used to solve the problem formulated by the authors. It also automatically proposes
an inference network that can be used for recognition. If the likelihood is Gaussian and p(x|z) is also Gaussian, one can directly marginalize x and work
with p(y|z) and p(z). The authors should at leas discuss the potential use of this method alongside with the BIGAN model which also provides a recognition model.

It is not clear how the HMC parameters are fixed.

The experiments do not have error bars (Figure 4.) This questions the significance of the results.

My overall impression is that there is little novelty in the proposed approach. Namely, using a GAN to learn the prior distribution, and then very well known
techniques to infer the original input image.

I have missed some references to related work on inverse problems. An example is:

https://arxiv.org/pdf/1712.03353.pdf


Is the original figure contained in the training set used to infer the GAN. If so that can lead to biased results.

I have missed a simple baseline in which one simply finds the training image that is closest to the corrupted observed or partially observed image.

My overall impression is that there is not much novelty in the paper as it is simply a combination of well known techniques. E.g. GANs and Bayesian inference with Monte Carlo methods.



**Experience Assessment:**

I have read many papers in this area.

**Review Assessment: Checking Correctness Of Derivations And Theory:**

I assessed the sensibility of the derivations and theory.

**Review Assessment: Checking Correctness Of Experiments:**

I assessed the sensibility of the experiments.

**Review Assessment: Thoroughness In Paper Reading:**

I read the paper at least twice and used my best judgement in assessing the paper.

---

> ### Author Response · Authors · 2019-11-09
> **Initial response to reviewer 1 -- Part (1/2)**
>
> Thank you for your valuable feedback! After carefully reading your comments we plan to modify the manuscript as discussed below (the planned changes are shown by ** … **). We would appreciate it if you could let us know whether the proposed changes address your concerns, or whether we have misinterpreted your comments.
> -----
> "Eq. (8) is expected to give very bad results. The reason is that it is very unlikely to sample from the prior configurations for z that are compatible with y."
>
> We agree that it is in general a bad idea to use this equation, especially when the likelihood is very informative. However, we have found that for likelihoods that are not very informative this method is still useful. Furthermore, we would like to point out that we do not use this equation for the results shown in the manuscript, but  rather use MCMC (eq. (9)) for all the results. ** We will include this discussion in the revised manuscript. **
> -----
> "The paper does not address learning any model parameters. e.g. the amount of noise."
>
> You are right, we do not learn parameters associated with the noise in this work. We do however learn the parameters associated with the forward model by mapping them back to the latent space of the GAN. This is most clear in the context of the physics-based model (Section 3.2), where we parameterize forward model using pixel-wise values of the initial temperature, and then learn these parameters using the proposed method.
>
> We note that the proposed approach can easily be extended to regime where likelihood is also unknown by incorporating likelihood-free inference methods like ABC or meta-learning approaches. ** We will include this discussion the revised version **.
> -----
> "A more principled approach would be to estimate the prior parameters using maximum likelihood estimation. That has already been done in the case of the variational autoencoder.
> The variational autoencoder is an already known method that can be used to solve the problem formulated by the authors. It also automatically proposes an inference network that can be used for recognition. If the likelihood is Gaussian and p(x|z) is also Gaussian, one can directly marginalize x and work with p(y|z) and p(z). The authors should at leas discuss the potential use of this method alongside with the BIGAN model which also provides a recognition model."
>
> We agree that we were lacking a proof that demonstrated the convergence of the proposed method for computing point estimates of the posterior. ** We have now derived this proof and will include it in the Appendix. ** In a nutshell, this proof establishes that with increasing the expressivity of the generator and the discriminator (increasing number weights) the point estimates computed using the proposed approach converges to the true point estimates of the posterior.
>
> We agree that using a variational autoencoder (VAE) in lieu of a GAN is an interesting extension of the proposed approach and that this can be accomplished in different ways. ** We are working on writing a concise description of these ideas and will include in the revised manuscript. ** However, we would like to point out that for image recovery tasks GANs have consistently demonstrated better performance than VAEs, as the latter tend to smear out images due to their maximum likelihood loss.
>
> While the idea of using corrupted images to train the VAE and inferring the latent variable, which would be the  - un-corrupted image is very interesting and using VAE with max. likelihood loss is an intriguing option, there are some major drawbacks of using it in the proposed Bayesian inference setting.
> •	It is well-known that image samples produced by VAEs are quite blurry and of poorer quality than GAN and fail to match the true data distribution. It is shown in earlier studies that they fail to match marginal distribution not only in visible space but also in latent space [1]. Since, the focus of our paper is to use these distributions as priors, we believe that it is better to select a model for these distributions and hence GAN is our preferred choice.
> •	Furthermore, VAEs are explicit density model and we have to select a model family (like Gaussian) for the latent variables. Therefore, in a setting where we treat un-corrupted images as latent variables (as suggested by the reviewer), and use max. likelihood as our loss function, we are forcing the latent variables to be close to that chosen family of distributions. This, might fail to capture complex inferred joint probability distribution seen in the examples considered in this manuscript, which is far from Gaussian (or any other simple distribution). It is also against the spirit of this work, where we want to make as few assumption as possible for our prior and use data to guide its final form.
>
> [1]. Rosca, M., Lakshminarayanan, B., & Mohamed, S. (2018). Distribution Matching in Variational Inference. ArXiv, abs/1802.06847.

---

> ### Author Response · Authors · 2019-11-09
> **Initial response to reviewer 1 -- Part (2/2)**
>
> "It is not clear how the HMC parameters are fixed."
>
> ** We will include a concise description of this in the revised version. **
> -----
> "The experiments do not have error bars (Figure 4.) This questions the significance of the results."
>
> ** This was an oversight on our part. We will include error bars in the revised version. **
> -----
> "My overall impression is that there is little novelty in the proposed approach. Namely, using a GAN to learn the prior distribution, and then very well known techniques to infer the original input image."
>
> We agree that the approach is simple; on the other hand, it is quite novel. We are not aware of any other work that performs uncertainty quantification using this combination of techniques in an unsupervised fashion. In contrast to other approaches [2, 3, 4], which use pairs of desired and measured images (x,y) to train the network, our approach only requires desired images (x). Furthermore, our work uniquely demonstrates the use of quantified uncertainty in active learning setup - finding the optimal location of sensors (successive measurement location) in an unsupervised fashion, which again is not reported before and has many potential applications.
> -----
> "I have missed some references to related work on inverse problems. An example is:
> https://arxiv.org/pdf/1712.03353.pdf"
>
> ** This is an interesting, related work. We will include it in the revised version, along with a description of how it relates to our approach. **
> -----
> "Is the original figure contained in the training set used to infer the GAN. If so that can lead to biased results."
>
> No, the original figures were not used in training the GAN in all examples. We studiously avoided this bias. ** We will mention this in the revised version.  **
> -----
> I have missed a simple baseline in which one simply finds the training image that is closest to the corrupted observed or partially observed image.
>
> ** We do not have this baseline but will include it in the revised version. **
> ---
>
> [2]. A. Kendall and Y. Gal, “What Uncertainties Do We Need in Bayesian Deep Learning for Computer Vision?”, NIPS (2017).
>  [3]. Kohl, S.A., Romera-Paredes, B., Meyer, C., Fauw, J.D., Ledsam, J.R., Maier-Hein, K.H., Eslami, S.M., Rezende, D.J., & Ronneberger, O. “A Probabilistic U-Net for Segmentation of Ambiguous Images”, NeurIPS (2018).
> [4]. Hu, S., Worrall, D., Knegt, S.J., Veeling, B., Huisman, H.J., & Welling, M., “Supervised Uncertainty Quantification for Segmentation with Multiple Annotations”. MICCAI (2019).

---

> ### Author Response · Authors · 2019-11-15
> **Final response to reviewer 1 -- Part (1/2)**
>
> Thank you for your valuable feedback! After carefully reviewing it, we have modified manuscript as discussed below (description of changes is enclosed within ** ... **).
>
> ----
> "Eq. (8) is expected to give very bad results. The reason is that it is very unlikely to sample from the prior configurations for z that are compatible with y."
>
> We agree that it is in general a bad idea to use this equation, especially when the likelihood is very informative. However, we have found that for likelihoods that are not very informative this method is still useful. Regardless, we would like to point out that we do not use this equation for the results shown in the manuscript, but  rather use MCMC (eq. (9)) for all the results.
> ** We have included this discussion in the revised manuscript in subsection 2.1. **
> -----
>
> "The paper does not address learning any model parameters. e.g. the amount of noise."
>
> You are right, we do not learn parameters associated with the noise in this work. We do however learn the parameters associated with the forward model by mapping them back to the latent space of the GAN. This is most clear in the context of the physics-based model (Section 3.2), where we parameterize forward model using pixel-wise values of the initial temperature, and then learn these parameters using the proposed method.
>
> We note that the proposed approach can easily be extended to regime where likelihood is also unknown by incorporating likelihood-free inference methods like ABC or meta-learning approaches.
> ** We have included this discussion in the Conclusion section of the revised version **.
> -----
> "A more principled approach would be to estimate the prior parameters using maximum likelihood estimation. That has already been done in the case of the variational autoencoder.
> The variational autoencoder is an already known method that can be used to solve the problem formulated by the authors. It also automatically proposes an inference network that can be used for recognition. If the likelihood is Gaussian and p(x|z) is also Gaussian, one can directly marginalize x and work with p(y|z) and p(z). The authors should at leas discuss the potential use of this method alongside with the BIGAN model which also provides a recognition model."
>
> We agree that we were lacking a proof that demonstrated the convergence of the proposed method for computing point estimates of the posterior.
> ** We have now derived this proof and  included it Appendix A**
> In a nutshell, this proof establishes that with increasing the expressivity of the generator and the discriminator (increasing number weights) the posterior density of the proposed method weakly converges to the true posterior density.
>
> We agree that using a variational autoencoder (VAE) in lieu of a GAN is an interesting extension of the proposed approach and that this can be accomplished in different ways.  However, we would like to point out that for image recovery tasks GANs have consistently demonstrated better performance than VAEs, as the latter tend to smear out images due to their maximum likelihood loss [1].
>
> While the idea of using corrupted images to train the VAE and inferring the latent variable, which would be the  - un-corrupted image is very interesting and using VAE with max. likelihood loss is an intriguing option, there are some major drawbacks of using it in the proposed Bayesian inference setting.
> • It is well-known that image samples produced by VAEs are quite blurry and of poorer quality than GAN and fail to match the true data distribution. It is shown in earlier studies that they fail to match marginal distribution not only in visible space but also in latent space [2]. Since, the focus of our paper is to use these distributions as priors, we believe that it is better to select a model for these distributions and hence GAN is our preferred choice.
> • Furthermore, VAEs are explicit density model and we have to select a model family (like Gaussian) for the latent variables. Therefore, in a setting where we treat un-corrupted images as latent variables (as suggested by the reviewer), and use max. likelihood as our loss function, we are forcing the latent variables to be close to that chosen family of distributions. This, might fail to capture complex inferred joint probability distribution seen in the examples considered in this manuscript, which is far from Gaussian (or any other simple distribution). It is also against the spirit of this work, where we want to make as few assumption as possible for our prior and use data to guide its final form.
>
> [1] R. A. Yeh, C. Chen, T. Yian Lim, A. G. Schwing, M. Hasegawa-Johnson, and M. N. Do, “Semantic image inpainting with deep generative models,” in Proceedings - 30th IEEE Conference on Computer Vision and Pattern Recognition, CVPR 2017, 2017, vol. 2017–January, pp. 6882–6890.
> [2]. Rosca, M., Lakshminarayanan, B., & Mohamed, S. (2018). Distribution Matching in Variational Inference. ArXiv, abs/1802.06847.

---

> > ### Author Response · Authors · 2019-11-15
> > **Final response to reviewer 1 -- Part (2/2)**
> >
> > "It is not clear how the HMC parameters are fixed."
> >
> > ** We have included concise description of this in section 3 of the revised version. **
> > -----
> > "The experiments do not have error bars (Figure 4.) This questions the significance of the results."
> >
> > ** This was an oversight on our part. We have included error bars in the revised version. **
> > -----
> > "My overall impression is that there is little novelty in the proposed approach. Namely, using a GAN to learn the prior distribution, and then very well known techniques to infer the original input image."
> >
> > We agree that the approach is simple; on the other hand, it is quite novel. We are not aware of any other work that performs uncertainty quantification using this combination of techniques in an unsupervised fashion. In contrast to other approaches [3, 4, 5], which use pairs of desired and measured images (x,y) to train the network, our approach only requires desired images (x). Furthermore, our work uniquely demonstrates the use of quantified uncertainty in active learning setup - finding the optimal location of sensors (successive measurement location) in an unsupervised fashion, which again is not reported before and has many potential applications.
> > **We have made this clear in a new subsection titled "Our contributions"**
> > -----
> > "I have missed some references to related work on inverse problems. An example is:
> > https://arxiv.org/pdf/1712.03353.pdf"
> >
> > ** This is an interesting, related work. We have included a reference to it in the revised version. **
> > -----
> > "Is the original figure contained in the training set used to infer the GAN. If so that can lead to biased results."
> >
> > No, the original figures were not used in training the GAN in all examples. We studiously avoided this bias.
> > ** We have mentioned in Section 3 of the revised version.  **
> > -----
> > I have missed a simple baseline in which one simply finds the training image that is closest to the corrupted observed or partially observed image.
> >
> > ** We have included this result (figure 10) in Appendix C. **
> > ---
> >
> > [3]. A. Kendall and Y. Gal, “What Uncertainties Do We Need in Bayesian Deep Learning for Computer Vision?”, NIPS (2017).
> >  [4]. Kohl, S.A., Romera-Paredes, B., Meyer, C., Fauw, J.D., Ledsam, J.R., Maier-Hein, K.H., Eslami, S.M., Rezende, D.J., & Ronneberger, O. “A Probabilistic U-Net for Segmentation of Ambiguous Images”, NeurIPS (2018).
> > [5]. Hu, S., Worrall, D., Knegt, S.J., Veeling, B., Huisman, H.J., & Welling, M., “Supervised Uncertainty Quantification for Segmentation with Multiple Annotations”. MICCAI (2019).

---

### Official Review · AnonReviewer2 · 2019-10-22
**Official Blind Review #2**

**Rating:** 3

**Review:**

The paper studies the Bayesian inferences with the generative adversarial network (GAN). In the first half of the paper, the general framework of the Bayes estimation is introduced. Then, The authors proposed how to incorporate GAN to the Bayesian inference. Some computational methods for calculating the mean of the statistic under the posterior distribution are described. Then, numerical experiments using MNIST and Celeb-A datasets are presented.

Though the Bayesian inference using GAN is a natural idea, learning algorithms proposed in this paper are simple and are not intensively developed. In numerical experiments, there is no comparison with major competitors besides random sampling in the active learning setup. Hence, the effectiveness and advantage of the proposed methods are not clear.
- In active learning, the proposed method should be compared with other methods such as Bayesian DNN using dropout, etc.
- How does the estimation accuracy of GAN relate to the estimation accuracy of the proposed method? Showing a quantitative description would be nice.


**Experience Assessment:**

I have read many papers in this area.

**Review Assessment: Checking Correctness Of Derivations And Theory:**

I assessed the sensibility of the derivations and theory.

**Review Assessment: Checking Correctness Of Experiments:**

I did not assess the experiments.

**Review Assessment: Thoroughness In Paper Reading:**

I made a quick assessment of this paper.

---

> ### Author Response · Authors · 2019-11-08
> **Initial response to reviewer 2**
>
> Thank you for your valuable feedback! After carefully reading it, we plan to modify the manuscript as discussed below (the planned changes are shown by ** … **). We would appreciate it if you could let us know whether the proposed changes address your concerns, or whether we have misinterpreted your comments.
>
> •	You have raised an interesting question about how the accuracy of the GAN impacts the accuracy of the proposed method. In order to address this, we have developed analytical estimates for the error in the point estimates computed using the proposed approach and show that these are intimately tied to error in computing the point estimates for the prior using the GAN. We have also demonstrated that as the generator and the discriminator of the GAN become more expressive this error tends to zero, and the exact point estimates, for both the prior and the posterior, are recovered. ** In the revised manuscript, we will include this mathematical analysis in the Appendix and refer to it in the main text. **
>
> •	We note that our method of inferring the desired image from the measured image is an unsupervised method; for training we only need a set of desired images  to construct the prior. We are not aware of any other unsupervised learning approach for solving these types of problems with quantified uncertainty. In that regard, the calculation of point-wise variance (our metric of uncertainty) is possible only using our approach, and therefore a direct comparison is not possible, since other supervised methods (explained below) cannot work in this setting where only set of desired images are available. ** We will clarify this unique aspect in the revised version of the manuscript. **
>
> •	There has been some work on computing the uncertainty in an inferred image within a supervised learning framework where pairs of measured and desired images are used for training the network [1, 2]. In these articles the authors have used methods like Bayesian dropout and variational autoencoder to compute uncertainty in the inferred images.  ** We will refer to these works in the revised version to better orient reader. **
>
>
> [1]. A. Kendall and Y. Gal, “What Uncertainties Do We Need in Bayesian Deep Learning for Computer Vision?”, NIPS (2017).
>  [2]. Kohl, S.A., Romera-Paredes, B., Meyer, C., Fauw, J.D., Ledsam, J.R., Maier-Hein, K.H., Eslami, S.M., Rezende, D.J., & Ronneberger, O. “A Probabilistic U-Net for Segmentation of Ambiguous Images”, NeurIPS (2018).

---

> ### Author Response · Authors · 2019-11-15
> **Final response to reviewer 2**
>
> Thank you for your valuable feedback! After carefully reviewing it, we have modified manuscript as discussed below (description of changes is enclosed within ** ... **).
>
> "Though the Bayesian inference using GAN is a natural idea, learning algorithms proposed in this paper are simple and are not intensively developed. In numerical experiments, there is no comparison with major competitors besides random sampling in the active learning setup. Hence, the effectiveness and advantage of the proposed methods are not clear."
>
> We have addressed this by responding to the specific questions below.
>
> "- In active learning, the proposed method should be compared with other methods such as Bayesian DNN using dropout, etc. "
>
> We are not aware of any other methods for computing uncertainty in recovered images that have been used to drive an active learning task in image inpainting. While methods based on dropout (\cite{Kendall2017a, Kendall2019}) or variational inference (\cite{Kohl2018a}) could be extended to accomplish this, this has not been done thus far.
> **We have added this comment in Section 3.1**
>
> Another big difference between the methods mentioned above and our approach is that while they require image pairs (true and corrupted images) for training, our approach only requires uncorrupted images. Thus while our algorithm relies on unsupervised learning, the other algorithms fall under the category of supervised learning.
> **We have also clarified this within the "Our Contributions" Section**
>
>
> "- How does the estimation accuracy of GAN relate to the estimation accuracy of the proposed method? Showing a quantitative description would be nice."
>
> We thank the reviewer for raising this important question.
> **We have addressed it thoroughly in Appendix A. We have provided a proof that demonstrates the weak convergence of the posterior density calculated using our method to the true posterior density as the number of weights in the discriminator and generator components of the GAN is increased.**

---

### Official Review · AnonReviewer3 · 2019-10-24
**Official Blind Review #3**

**Rating:** 3

**Review:**

This paper proposes to use a trained GAN model as the prior distribution for Bayesian inference to quantify the uncertainty. As for me, the best application of this paper is to restore a corrupted image, which shares a lot of common properties in image restoration, denoising and image reconstruction. I do like the extension of applying the idea in physics problems. And the results demonstrate at some extent, the proposed method could evaluate some uncertainty.

The idea is pretty simple and the paper is easy to read.  Nonetheless, there are some issues:

A big issue of this paper is the deviation of purpose and method. As the paper claims to quantify the uncertainty, the paper is supposed to give specific quantitative metric or values to probe the uncertainty. However, the paper demonstrates to us only the ability, not exactly “quantification”. I’d like to see a specific metric of uncertainty that could only be calculated through the proposed method.

There are some grammar issues in the paper. For example. “…we the MAP…” in the 7th page.

Given my major issue seems to be quite problematic, I currently would weakly reject this paper. But I don’t have a full picture over this area, I’ll read the rebuttal and see if I could raise the score.

**Experience Assessment:**

I have published one or two papers in this area.

**Review Assessment: Checking Correctness Of Derivations And Theory:**

I assessed the sensibility of the derivations and theory.

**Review Assessment: Checking Correctness Of Experiments:**

I assessed the sensibility of the experiments.

**Review Assessment: Thoroughness In Paper Reading:**

I read the paper at least twice and used my best judgement in assessing the paper.

---

> ### Author Response · Authors · 2019-11-08
> **Initial response to reviewer 3**
>
> Thank you for your valuable feedback! After carefully reviewing it, we plan to modify the manuscript as discussed below (the planned changes are shown by ** ... **). We would appreciate it if you could let us know whether the proposed changes address your concerns, or whether we have misinterpreted your comments.
>
> "A big issue of this paper is the deviation of purpose and method. As the paper claims to quantify the uncertainty, the paper is supposed to give specific quantitative metric or values to probe the uncertainty. However, the paper demonstrates to us only the ability, not exactly 'quantification'. I’d like to see a specific metric of uncertainty that could only be calculated through the proposed method."
>
> You are right, the main purpose of the method is to quantify uncertainty in the task of  image inference. Given this, we treat the inference as a stochastic problem, and develop an expression for the probability density function of the inferred image (i.e. joint pdf for each pixel of the inferred image). Once this is done, we sample from this distribution and compute any appropriate point estimate that can quantify the uncertainty in the inference. In our work, we have chosen the "pixel-wise" variance as this metric. Note that this metric is a field and not a scalar quantity and is plotted as an image. We have computed this metric for every example in the manuscript (see last row of figure 2, 3, 5 etc). ** However, we have been remiss in not highlighting, or bringing the reader’s attention to it. In the revised version of the paper we will do this. **
>
> Some more things to note:
> 1.	In one example (Figure 4) we compute a scalar metric (that is the average variance/per pixel over the entire image) for the  inferred images, and show that this measure increases with increasing noise in the input, as it should. ** In the revised version, we will draw the reader's attention to this example. **
>
> 2. 	We note that our method of inferring the desired image from the measured image is an unsupervised method; in that for training we only need a set of desired images to construct the prior. We are not aware of any other unsupervised learning approach for solving these types of problems with quantified uncertainty. In that regards, the calculation of pixel-wise variance (our metric of uncertainty) in an unsupervised setting is possible only using our approach. ** We will clarify this unique aspect in the revised version of the manuscript. **
>
> 3.	We note that there has been recent work on computing the uncertainty in an inferred image within a supervised learning framework where pairs of measured and desired images are used for training the network. In these articles the authors have used methods like Bayesian dropout to compute uncertainty in the inferred images [1]. Similar to what we have done, these authors have also plotted the point-wise variance as a quantitative metric of uncertainty. ** We will refer to these works in the revised version to better orient the readers. **
>
> 4.	We note that we have gone beyond just computing the metric of uncertainty (point-wise variance) and also described how it might be useful in making the subsequent measurement in the context of an active learning approach, which to the best of our knowledge has not been done previously in Bayesian deep learning applied to image inference.
>
>
> "There are some grammar issues in the paper. For example. '…we the MAP…' in the 7th page."
>
> ** We are doing a through scrub of manuscript in order to catch these.  **
>
>
> [1]. A. Kendall and Y. Gal, “What Uncertainties Do We Need in Bayesian Deep Learning for Computer Vision?”, NIPS (2017).

---

> ### Author Response · Authors · 2019-11-15
> **Final response to reviewer 3**
>
> Thank you for your valuable and constructive feedback! After carefully reviewing it, we have modified the manuscript as discussed below (description of changes is enclosed within ** ... **).
>
> "A big issue of this paper is the deviation of purpose and method. As the paper claims to quantify the uncertainty, the paper is supposed to give specific quantitative metric or values to probe the uncertainty. However, the paper demonstrates to us only the ability, not exactly 'quantification'. I’d like to see a specific metric of uncertainty that could only be calculated through the proposed method."
>
> You are right, the main purpose of the method is to quantify uncertainty in the task of  image inference. In fact, to our knowledge the method we describe is the only unsupervised learning method for quantifying uncertainty in a deep-learning based image-recovery task. We treat the inference as a stochastic problem, and develop an expression for the probability density function of the inferred image (i.e. joint pdf for each pixel of the inferred image). Once this is done, we sample from this distribution and compute any appropriate point estimate that can quantify the uncertainty in the inference. In our work, we have chosen the "pixel-wise" variance as this metric. Note that this metric is a field and not a scalar quantity and is plotted as an image. We have computed this metric for every example in the manuscript (see last row of figure 2, 3, 5 etc).
> ** However, we have been remiss in not highlighting, or bringing the reader’s attention to it. In the revised version of the paper we have done this by highlighting this field in the images and its description in the text. **
>
> Some more things to note:
> 1. In one example (Figure 4) we compute a scalar metric (that is the average variance/per pixel over the entire image) for the  inferred images, and show that this measure increases with increasing noise in the input, as it should.
> ** In the revised version, we have drawn the reader's attention to this example. **
>
> 2. We note that our method of inferring the desired image from the measured image is an unsupervised method; in that for training we only need a set of desired images to construct the prior. We are not aware of any other unsupervised learning approach for solving these types of problems with quantified uncertainty. In that regards, the calculation of pixel-wise variance (our metric of uncertainty) in an unsupervised setting is possible only using our approach.
> ** We have clarified this unique aspect in the revised version of the manuscript by listing it under a new subsection titled "Our Contribution" **
>
> 3. We note that there has been recent work on computing the uncertainty in an inferred image within a supervised learning framework where pairs of measured and desired images are used for training the network. In these articles the authors have used methods like Bayesian dropout to compute uncertainty in the inferred images [1]. Similar to what we have done, these authors have also plotted the point-wise variance as a quantitative metric of uncertainty.
> ** We have referred to these works in the "Related Work" subsection of the revised version of the manuscript. **
>
> 4. We note that we have gone beyond just computing the metric of uncertainty (point-wise variance) and also described how it might be useful in making the subsequent measurement in the context of an active learning approach, which to the best of our knowledge has not been done previously in Bayesian deep learning applied to image inference.
> ** We have clarified this unique aspect in the revised version of the manuscript by listing it under a new subsection titled "Our Contribution" **
>
> "There are some grammar issues in the paper. For example. '…we the MAP…' in the 7th page."
>
> ** We have done a through scrub of manuscript in order to catch these.  **
>
>
> [1]. A. Kendall and Y. Gal, “What Uncertainties Do We Need in Bayesian Deep Learning for Computer Vision?”, NIPS (2017).

---

### Author Response · Authors · 2019-11-15
**tl;dr**

Based on all the reviewer's response we have recognized that we were remiss in not clarifying our main contributions in the manuscript. We have done so in the revised version and repeat them below.

The main contribution of this paper can be summarized as follows:
•	A novel method for performing Bayesian inference involving complex priors and high dimensional posterior. In the proposed method we utilize the distribution learned by a GAN as a surrogate for the prior distribution and reformulate the inference problem in the low-dimensional latent space of the GAN.

•	A theoretical analysis of the weak convergence of the posterior density learned by the proposed method to the true posterior density.

•	 Novel unsupervised image denoising and inpainting algorithms with quantitative measures of uncertainty through pixel-wise variance.

•	Application of the proposed method to physics-based inference problems.

•	Demonstration of the utility of uncertainty quantification to facilitate active learning.

---

### Decision · Program_Chairs · 2019-12-19

**Decision:**

Reject

**Comment:**

This paper suggests a Bayesian approach to make inference about latent variables for image inference tasks. While the idea in the paper seems elegant and simple, reviewers pointed out a few concerns, including lack of comparisons, missing references, and requested for more extensive validations. While a few comments might have been misunderstandings (eg lack of quantification - seems to be resolved by author’s comments), other comments are not (eg equation (8) needs further justification even if the final results don’t use it). We encourage authors to carefully review comments and edit the manuscript (perhaps some appendix items should be in the main to reduce confusion) for resubmitting to future conferences.